# Association of Multiple Cardiovascular Risk Factors with Musculoskeletal Function in Acute Coronary Syndrome Ward Inpatients

**DOI:** 10.3390/healthcare11070954

**Published:** 2023-03-27

**Authors:** Gabriel Parisotto, Luis Felipe Fonseca Reis, Mauricio Sant’Anna Junior, Jannis Papathanasiou, Agnaldo José Lopes, Arthur Sá Ferreira

**Affiliations:** 1Postgraduate Program of Rehabilitation Sciences, Augusto Motta University Center/UNISUAM, Rio de Janeiro 21032-060, Brazil; gabriel_parizoto@yahoo.com.br (G.P.); luisfelipefreis@gmail.com (L.F.F.R.); alopes@souunisuam.com.br (A.J.L.); 2Department of Physical Therapy, Rio de Janeiro Federal Institute, Rio de Janeiro 21710-240, Brazil; mauricio.junior@ifrj.edu.br; 3Department of Medical Imaging, Allergology & Physiotherapy, Faculty of Dental Medicine, Medical University of Plovdiv, 4002 Plovdiv, Bulgaria; giannipap@yahoo.co.uk; 4Department of Kinesitherapy, Faculty of Public Health, Medical University of Sofia, 1431 Sofia, Bulgaria

**Keywords:** coronary care units, muscle strength, rehabilitation, risk factors

## Abstract

This study explored the association of multiple risk factors with musculoskeletal function in adults hospitalized for acute coronary syndrome. Sixty-nine inpatients (55 ± 6 years; 67% male) admitted to the cardiology ward within <12 h were assessed regarding stress, smoking, alcoholism, hypertension, diabetes mellitus, and obesity. The musculoskeletal function was assessed by predicted values of handgrip strength of the dominant hand (HGS-D_%_) and maximal inspiratory and expiratory pressures (MIP_%_ and MEP_%_, respectively). After adjustment by age and sex, drinking habits showed the strongest linear association with the total number of cardiovascular disease risk factors [standardized ß, *p*-value] (ß = 0.110, *p* < 0.001), followed by smoking load (ß = 0.028, *p* = 0.009). Associations were also observed for HGS-D_%_ with mean blood pressure (ß = 0.019 [0.001; 0.037], *p* = 0.048); MIP_%_ with mean blood pressure (ß = 0.025 [0.006; 0.043], *p* = 0.013); and MEP_%_ with drinking habits (ß = 0.009 [0.002; 0.016], *p* = 0.013) and body mass index (ß = 0.008 [0.000; 0.015], *p* = 0.035). Peripheral and respiratory muscle strength must be interpreted in the context of its association with cardiovascular disease risk factors in adults hospitalized for acute coronary syndrome.

## 1. Introduction

Cardiovascular diseases (CVD) are a major burden for public health worldwide despite a decreasing trend in prevalence over the last three decades [1]. The majority of deaths from CVD are associated with multiple, either modifiable (e.g., health behaviors and some comorbidities) or non-modifiable (e.g., age, sex, and genetics) risk factors [2]. Modifiable risk factors constitute the main therapeutic targets for intervention and rehabilitation programs via the management of health behaviors, including controlling weight, reducing stress, stopping smoking and drinking habits, and promoting physical activity [2]. Controlling comorbidities such as hypertension, diabetes mellitus, and obesity is also advocated to increase both the expectation and quality of life in this population [2]. Recent data suggest the Brazilian population follows the global trend, with a reduction in the risk of death from CVD compared to the 1990s [3]. However, with the recent flattening of the rate of decline in CVD mortality in Brazil, the research on CVD risk factors and their functional repercussion gains additional momentum [4].

Despite the promising reduction in the overall burden of CVD, ischemic heart disease ranks as the 2nd leading cause of disability-adjusted life-years after a 17% increase from 2007–2017 globally and the 1st leading cause of years of life lost in Brazil [3]. In recent years, an increasing rate of hospitalization due to acute coronary syndrome (ACS) has been reported, the latter being responsible for a significant increase in costs for individuals and both public and private healthcare systems [5]. Hospitalizations for ACS are associated with several factors, including health behaviors (e.g., drinking habits, psychological factors) and comorbidities (e.g., uncontrolled hypertension) [6].

Skeletal muscle weakness is one of the impairments that make up the pathological condition as a result of ACS [7]. Adults with ACS undergoing elective surgery have a significant impairment in respiratory muscle strength at preoperative evaluation [8]. Muscle weakness is also a known complication acquired by critically ill patients that have associated with poor outcomes such as longer length of stay and more days of mechanical ventilation [9]. The relationship between skeletal muscle function and CVD risk (e.g., smoking [10]) or comorbidities (e.g., hypertension [11], diabetes mellitus [12], obesity [13,14,15]) has been investigated in several populations and a variety of methods but remains controversial. In adults with ACS, reports on the association between musculoskeletal function and CVD risk are also controversial and are difficult to compare due to major differences in study design (cross-sectional [16,17,18,19]; longitudinal [20,21]; Mendelian randomization [22,23]) and populations (men and women separately [16]; men only [17]; women only [18]; general population in a surgical intensive care unit [20]; cases of coronary artery disease/myocardial infarction and controls [21,22,23]). Particularly, to what extent musculoskeletal function (i.e., peripheral or respiratory muscle strength) is associated with comorbidities and lifestyle—either as independent CVD risk factors or a clustered group of comorbidities—in patients admitted in the cardiology ward for ACS remains unknown. Understanding this relationship might allow tailoring patient-centered interventions for cardiac and pulmonary rehabilitation. Therefore, this study explored the association of multiple CVD risk factors with musculoskeletal function in adults hospitalized for ACS.

## 2. Materials and Methods

### 2.1. Study Design and Ethics

This is a primary, cross-sectional study. The study protocol was approved (No. 19634419.2.0000.5235) by the Institutional Ethics Committee before its execution according to national regulations (National Health Council Resolution No. 466/2012) and the World Medical Association Declaration of Helsinki as revised in 2013. Participants signed an informed consent form after being informed about the study’s aims, design, and protocol. This study is reported following the Strengthening the Reporting of Observational Studies in Epidemiology [24].

### 2.2. Setting and Participants

Data collection was conducted in a cardiology ward of a primary-to-secondary hospital (Hospital Dr. Wilson Franco Rodrigues, Roraima State, Brazil) between September 2019 and September 2020. All assessments were performed by the same examiner.

The inclusion criteria for the study were: <12 h of hospital admission; age between 18 and 60 years (to limit the effects of aging and/or sarcopenia); being cooperative, breathing room air without the use of endotracheal prosthesis; and medical diagnosis of ACS confirmed by medical records. Participants were not included if they presented with invasive mechanical ventilation, pregnancy, Glasgow score < 11, Global Registry of Acute Coronary Events (GRACE) risk score > 140 (high risk for in-hospital death), or at least one of the following conditions: abdominal distension, ascites, neuromuscular diseases (Guillain Barrett syndrome, amyotrophic lateral sclerosis, amyotrophy, myasthenia gravis, polymyositis), respiratory diseases (chronic obstructive pulmonary disease, asthma, pulmonary fibrosis), hemoglobin < 10 g/dL, tachycardia (>140 bpm), chest pain, or palpitation.

### 2.3. Clinical Assessment

The enrolled participants underwent anamnesis with a standard case report form to collect data regarding clinical status, health behaviors, comorbidities, and musculoskeletal function. The total length of stay was determined from ward admission until hospital discharge.

Stress was evaluated by the Portuguese-Brazil version of the Stress Symptom Inventory for Adults (ISSL) [25]. The ISSL assesses the stress level grouped as alertness (last 24 h), resistance (last week), or exhaustion (last month). The validity of the ISSL is high (Cronbach alpha = 0.912).

Smoking was evaluated by the smoking load, calculated as average packs smoked per day multiplied by the duration of smoking in years [26].

Alcohol drinking habits were evaluated by the Portuguese-Brazil version of the Alcohol Use Problems Identification Test (AUDIT) [27]. The AUDIT consists of 10 questions referring to the last 12 months of consumption; the first three questions measure the amount and frequency of regular or occasional use of alcohol, the following three questions investigate symptoms of addiction, and the remaining four are about recent problems in life related to alcohol consumption. The score ranges from 0 points (low risk of alcohol dependence) to 40 points (probable alcohol dependence). The psychometric characteristics of the AUDIT showed excellent reliability (Intraclass Correlation Coefficient = 0.80, Cronbach α = 0.81).

Blood pressure was measured using a validated digital device Morefitness M/F—390, following the international protocol [28]; pulse pressure and mean blood pressure (MBP) were calculated. Heart rate and peripheral perfusion were assessed by a portable pulse oximeter Intermed Model SAT-200 (Contec Medical Systems) with a digital sensor.

Plasma glucose was measured using Accu-Chek Softclix lancet and Accu-Chek Active blood glucose monitor (Roche Diagnostics) following the international protocol.

Body mass was measured at the bedside in the standing posture, barefoot, and wearing just a short using a portable digital scale G-TECH Glass 10 (Accumed Products Hospital Medical Ltda., Rio de Janeiro, Brazil). Body height was measured from the top of the head to the sole using a flexible tape (Company Vonder) at the bedside in the supine position, with the bed in a fully horizontal position. Body mass and height were used to calculate the body mass index (BMI).

Body composition was assessed using a bioimpedance analyzer Biodynamics BIA 310 (Biodynamics Corporation; accuracy: 0.1%, frequency 50 kHz); sex, age, body height, and body mass data were used to estimate body fat mass, body fat percentage, body lean mass, basal metabolic rate, and hydration status. Abdominal circumference was assessed using a flexible measuring tape positioned at the umbilical line taking as a reference the midpoint of the iliac crest and the last rib at the end of the expiratory phase with the patient in the supine position [29].

### 2.4. Assessment of Cardiovascular Disease Risk Factors

The number of CVD risk factors for each participant comprised the sum of all CVD risk factors after dichotomization (present = 1, absent = 0) using pre-established cut-off values:Stress: ISSL score ≥ 7 for alertness, ≥4 for resistance, or ≥7 for exhaustion [25];Smoking: smoking load ≥ 10 pack-years as a cut-off point related to the impact of smoking on lung function [30];Alcohol drinking: AUDIT score ≥ 8 for men and ≥5 for women [31];Hypertension: systolic blood pressure ≥ 140 mmHg and/or diastolic blood pressure ≥ 90 mmHg, or the use of antihypertensive drugs [28];Diabetes mellitus: fasting glycemia ≥ 126 mg/dl, or the use of hypoglycemic drugs [32];Obesity: BMI ≥ 30 kg/m^2^ [33].

### 2.5. Assessment of Musculoskeletal Function

Peripheral muscle strength was assessed by measurement of handgrip strength of the dominant hand (HGS-D) using an analog dynamometer (Instrutherm Instrumentos de Medição LTDA) according to the American Association of Hand Therapists [34]. Three tests were performed with a 1-min interval between tests; the highest reference value was used to increase the validity and reliability. Predicted values (HGS-D_%_) were estimated from a national reference equation and used in subsequent analyses [35].

Respiratory muscle strength was assessed by measuring maximum inspiratory and expiratory pressures (absolute values of MIP and MEP, respectively) following the recommendations of the American Thoracic Society [36] using a calibrated manovacuometer MRN 020002 (Murenas Produtos para Saúde LTDA). The mouthpiece was placed firmly in the patient’s mouth, and the escape orifice was unobstructed, as it has the function of keeping the glottis open and thus preventing the action of the oropharyngeal facial musculature that can alter the results. The obstruction valve remained open when reaching residual volume and total lung capacity and was obstructed at the time of the evaluation. The number of evaluations performed consisted of three consecutive measurements, obtaining the highest value among them, lasting 2 s of MIP and MEP without leaks. The predicted values (MIP_%_ and MEP_%_) were estimated from national reference equations and used in subsequent analyses [37].

### 2.6. Statistical Methods

The sample size was determined a priori using G*Power 3.1 software. For a linear multiple regression analysis (H_0_: R^2^ = 0), given a 5% type-I error, 20% type-II error, 6 predictors, and a correlation between the outcome and predictors of 0.2 (overall *ρ*^2^ = 0.24), a minimum of 50 participants is required.

Analysis was performed using R project 4.0.2 after importing data typed into an electronic spreadsheet in Excel (Microsoft, Redmond, WA, USA). The statistical significance value is set to *p* < 0.05 (two-tailed). A complete-case analysis was conducted, as there were no missing values for the study outcomes.

Values are shown as mean ± SD for continuous variables, whereas categorical variables are described as a frequency (%); boxplots were generated for visualization of both summary of distributions and the relationship between variables. The Shapiro-Wilk test and histogram analysis were used to check the normality of the variables. Generalized (first-order) linear models were used for regression due to a nonnormal distribution of most variables; hence, quantitative variables were not log-transformed. Initially, the association between multiple CVD risk factors and each of their surrogate measures was explored (ISSL sumscore, MBP, glycemia, and BMI with Gamma family and log link; smoking load and AUDIT score with negative binomial family and log link). In sequence, the association between a surrogate measure of each CVD risk factor (ISSL sumscore, smoking load, AUDIT, MBP, glycemia, and BMI) and musculoskeletal function was explored (HGS-D_%_, MIP_%_, and MEP_%_; all with a Gamma family and log link). Each fixed-effects model includes adjustments for age and sex as they comprise covariates shared among the selected outcomes. The raw and standardized regression coefficients (β) with respective confidence intervals (95%CI), *p* values, and coefficient of determination *R*^2^ are reported.

## 3. Results

A total of 76 participants were screened for eligibility; six were excluded due to missing data, and one declined consent. Table 1 shows the demographic and clinical characteristics of the sample. Sixty-nine inpatients at the cardiology ward were enrolled (age 55 ± 6 years), and most participants were men (n = 46, 67% male). Most participants (n = 19, 24%) presented four CVD risk factors; the frequency of CVD risk factors was smaller for both a higher—5 (n = 6, 9%)—and a lower number of multiple CVD risk factors—3 (n = 17, 25%), 2 (n = 16, 23%) or 1 (n = 11, 16%). Stress was the most frequent health behavior (n = 43, 62%), followed by smoking (n = 30, 43%) and drinking habits (n = 21, 30%). The most common comorbidity was hypertension (n = 60, 87%), followed by obesity (n = 31, 45%) and diabetes mellitus (n = 15, 22%). At cardiology ward admission, GRACE score averaged 107 ± 23; evidence of peripheral (HGS-D_%_ = 74 ± 21%) and respiratory musculoskeletal weakness were observed (MIP_%_ = 65 ± 27%, MEP_%_ = 57 ± 22%).

The association analysis between the number of CVD risk factors and each CVD risk factor is shown in Table 2 and Figure 1. After adjustment by age and sex, the AUDIT score showed the strongest linear association with the number of CVD risk factors [standardized ß, *p*-value] (ß = 0.110, *p* < 0.001), followed by smoking load (ß = 0.028, *p* = 0.009), ISSL sumscore (ß = 0.021, *p* = 0.008), BMI (ß = 0.016, *p* < 0.001), and glycemia (ß = 0.002, *p* = 0.047). No evidence of a linear association was observed between MBP and the number of CVD risk factors (ß = 0.000, *p* = 0.994).

The association analysis between CVD risk factors and musculoskeletal function is shown in Table 3. After adjustment by age and sex, evidence of a direct linear association was observed between HGS-D_%_ and MBP (ß = 0.019 [0.001; 0.037], *p* = 0.048). Assuming all other variables are constant, a mean change of 0.7% in HGS-D_%_ is expected for a 1-mmHg change in MBP. Likewise, evidence of a direct linear association was observed between MIP_%_ and MBP (ß = 0.025 [0.006; 0.043], *p* = 0.013). Assuming all other variables are constant, a mean change of 1.3% in MIP_%_ is expected with a 1-mmHg change in MBP. Finally, evidence of a direct linear association was observed between MEP_%_ and AUDIT score (ß = 0.009 [0.002; 0.016], *p* = 0.013) and BMI (ß = 0.008 [0.000; 0.015], *p* = 0.035). Assuming all other variables are constant, a mean change of 2% or 2.5% in MEP_%_ is expected with a 1-point change in AUDIT sumscore or a 1-kg/m^2^ change in BMI, respectively.

## 4. Discussion

This study explored the association of multiple CVD risk factors with musculoskeletal function in adults hospitalized for ACS. The major findings suggest that peripheral muscle strength is directly associated with mean blood pressure, whereas respiratory muscle strength is directly associated with mean blood pressure, alcohol drinking, and body mass index in adults hospitalized for ACS. Major strengths comprise using valid, reliable instruments to evaluate the CVD risk factors and respiratory and peripheral muscle strength. In addition, the demographic and risk factor profile of this cohort is similar to other studies in adults hospitalized for ACS [38,39], highlighting the external validity of our findings.

Peripheral muscle strength showed evidence of association with all surrogate measures of CVD risk factors (ISSL sumscore, smoking load, AUDIT score, glycemia, BMI) but MBP. Reports on the association between HGS and CVD risk are also controversial and are difficult to compare herein due to major differences in designs and populations. Cross-sectional studies conducted in the general population reported an inverse association between HGS/BMI and systolic blood pressure among United States adults (men and women separately [16]), in Taiwan adults (men only [17]), and Japanese adults (women only [18]). Another cross-sectional study in Chinese elderlies found a low discriminative power of HGS/BMI (or HGS/weight) on several risk factors for CVD—including hypertension and diabetes mellitus—as well as to presenting ≥ 1 CVD risk factor [19]. A longitudinal analysis of patients admitted to a general population surgical intensive care unit reported Medical Research Council scores but not HGS-predicted length of stay, but only 1 (0.9%) patient had ACS as an admission diagnosis [20]. Another longitudinal study showed a reduced mortality risk with higher HGS univariately and after adjustment for age, gender, and other cardiovascular risk factors incusing BMI, type-2 diabetes mellitus, hypertension, and a history of smoking [21]. A Mendelian randomization study of cases with coronary artery disease/myocardial infarction and controls showed that each 1-kg increase in HGS decreased CAD risk by 6%; the study also reported no significant association was found for type 2 diabetes, BMI, and fasting glucose [22]. Another Mendelian randomization study showed that both observational and genetically predicted low handgrip strength was associated with high all-cause and particularly cardiovascular mortality after adjustment for age, sex, phenotypes for diabetes, and body mass index, among others [23]. Discrepancies among these findings might also be explained by different methods being used, e.g., ‘absolute’ or ‘relative’ handgrip strength. However, the findings of this study using prediction equations for HGS did not support such discrepancies.

Respiratory muscle strength was found to be positively associated with some surrogate measures of CVD risk factors (MBP, AUDIT score, and BMI). Such a relationship appears paradoxical as smoking [10], hypertension [11], and diabetes mellitus [12] have been inversely related to respiratory muscle function. The relationship with obesity seems still debatable; MIP and MEP were reported as not different across classifications of nutritional status by BMI [13] and positively correlated with BMI [14,15]. The relationship of respiratory muscle strength with stress or alcohol drinking remains not investigated, although evidence points to a direct association between either risk factors and mortality [40,41]. It is worth noting that such relationships were reported independently for each CVD risk factor in different populations, using self-reported, non-standard, or different instruments for assessing CVD risk factors; and analyzing raw and/or predicting functional outcomes with or without adjustment for possible covariates. The reasoning underlying the above-mentioned direct association remains uncertain. Given that CVD is multifactorial, the interactive, complex nature of their risk factors has been investigated in adults with ACS [42,43]. Our findings corroborate this interactive nature as all (ISSL sumscore, smoking load, AUDIT score, glycemia, BMI) but one (MBP) surrogate measure of cardiovascular risk factors was positively associated with the total number of CVD risk factors (Table 2). Such lack of association of MPB may be explained by the majority of patients (n = 60, 87%) reporting hypertension, which is a leading cause of ACS. Accordingly, the majority of participants (28%) showed four risk factors, similar to other cohorts reported in Brazil [38] and other countries [39].

The major limitation of this study is the cross-sectional design that precludes inference about the cause-effect associations for the observed models. Another major limitation is that CVD risk factors were operationalized with a single measure, e.g., a point measurement of blood pressure and fasting glycemia, that might express the current condition rather than the trait for a risk profile. In addition, data on additional CVD risk factors (e.g., physical activity) and functional outcomes (e.g., pulmonary function, functional exercise capacity) were not collected and might provide additional insights into the relationships investigated herein. For instance, to what extent are inspiratory and expiratory pressures correlated with smoking and not necessarily muscle strength—and thus are more correlated with pulmonary diseases than CAD itself—should be further investigated. Moreover, the possible confounding effect of drug treatment other than antihypertensive or hypoglycemic agents was not considered. Finally, even though national prediction equations were used, nationwide regional differences—mainly on anthropometry and nutritional status—might help explain the low predicted values for the musculoskeletal functions observed herein.

The clinical implications of the current study findings comprise that knowing the CVD risk profile might allow planning interventions for adults with ACS while staying in a cardiology ward. With new evidence suggesting a causal relationship between HGS and all-cause and cardiovascular mortality [22,23], it is expected that these findings might also contribute to primary-to-secondary prevention for further guidance on changing health behaviors and controlling for comorbidities that can lead to an increase in both life expectancy and quality of life in this population as well as to avoid hospital readmissions [2]. Future studies might investigate whether biopsychosocial factors have moderation and/or mediation effects that help explain the causal association between surrogate measures of CVD risk factors and muscle strength in adults hospitalized for ACS. In addition, whether peripheral or respiratory muscle strengths are associated with (or can predict) barriers to participating in cardiac rehabilitation programs (e.g., comorbidities/functional status, perceived needs, personal/family problems, travel/work conflicts, and access) after hospital discharge requires further investigation in this population.

## 5. Conclusions

Peripheral muscle strength is directly associated with mean blood pressure, whereas respiratory muscle strength is directly associated with mean blood pressure, alcohol drinking, and body mass index in adults hospitalized for ACS. These findings reinforce the association of skeletal muscle functional status with comorbidities (e.g., hypertension, obesity) and health behaviors (drinking habits). Skeletal muscle strength measured at cardiology ward admission should be interpreted in the context of its association with CVD risk factors in this population.

## Figures and Tables

**Figure 1 healthcare-11-00954-f001:**
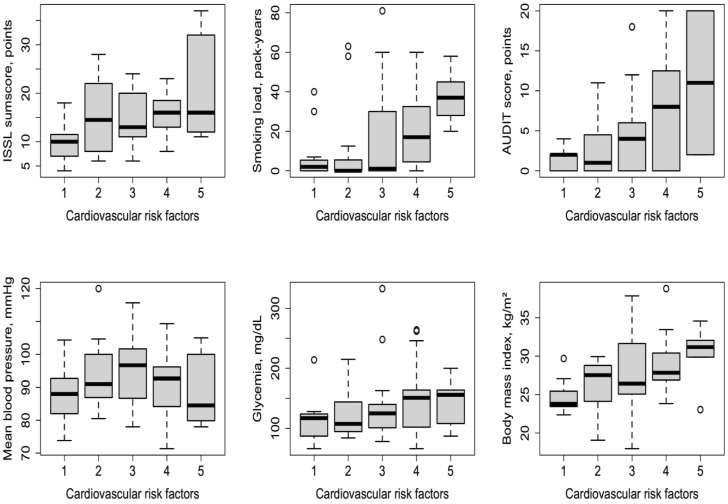
Boxplots of independent surrogate measures of cardiovascular disease risk factors (ISSL sumscore, smoking load, AUDIT score, mean blood pressure, glycemia, body mass index) across the combination of risk factors (stress, smoking habits, alcohol drinking, hypertension, diabetes mellitus, obesity). The horizontal axis represents the total number of cardiovascular risk factors (1–5) identified per participant based on the clinical and/or laboratory assessment. Notice no patient presented with all six cardiovascular risk factors simultaneously. Circles represent outliers.

**Table 1 healthcare-11-00954-t001:** Descriptive analysis of the sample of patients hospitalized for acute coronary syndrome in the cardiology ward.

Variable	Description	Values
Sample size, n (%)		69
	Female	23 (33%)
	Male	46 (67%)
Age, years		55 ± 6
GRACE risk score, n (%)		107 ± 23
	Low risk	32 (46%)
	Intermediate risk	24 (35%)
	High risk	13 (19%)
Clinical/laboratory exams		
	Heart rate, b/min	74 ± 12
	Systolic blood pressure, mmHg	122 ± 16
	Diastolic blood pressure, mmHg	76 ± 10
	Pulse pressure, mmHg	46 ± 15
	Mean pressure, mmHg	92 ± 10
	Blood saturation, %	97 ± 1
	Glycemia, mg/dL	135 ± 54
Anthropometry		
	Body height, m	1.62 ± 0.09
	Body mass, kg	72.0 ± 12.8
	Body mass index, kg/m^2^	27.4 ± 3.9
	Abdominal circumference, cm	97.0 ± 12.2
Nutritional status, n (%)		
	Thin	1 (1%)
	Eutrophic	20 (29%)
	Overweight	33 (48%)
	Obese I	13 (19%)
	Obese II	2 (3%)
Body composition		
	Body fat, %	21 ± 7
	Fat mass, kg	24 ± 9
	Thin mass, kg	48 ± 10
Health behaviors		
Smoking load, pack-years		16.1 ± 21.0
AUDIT score, n (%)		
	Probable dependency	3 (4%)
	High risk	4 (6%)
	Medium risk	13 (19%)
	Low risk	49 (71%)
ISSL, Phase I, n (%)		
	Alert	13 (19%)
	No alert	56 (81%)
ISSL, Phase II, n (%)		
	Resistant	42 (61%)
	No resistant	27 (39%)
ISSL, Phase III, n (%)		
	Exhaustion	13 (19%)
	No exhaustion	56 (81%)
Risk factors for cardiovascular disease, n (%)		
	1	11 (16%)
	2	16 (23%)
	3	17 (25%)
	4	19 (28%)
	5	6 (9%)
Risk factors, n (%)		
	Hypertension	60 (87%)
	Stress	43 (62%)
	Obesity	31 (45%)
	Smoking	30 (43%)
	Drinking	21 (30%)
	Diabetes mellitus	15 (22%)
Length of stay, days		40 ± 26
Musculoskeletal function		
Handgrip strength		
	Dominant hand, kg	29 ± 10
	Dominant hand, predict %	74 ± 21
Respiratory muscle strength		
	Maximal inspiratory pressure, cmH_2_O	−67 ± 31
	Maximal expiratory pressure, cmH_2_O	61 ± 28
	Maximal inspiratory pressure, predict %	65 ± 27
	Maximal expiratory pressure, predict %	57 ± 22

GRACE: Global Registry of Acute Coronary Events. AUDIT: Alcohol Use Problems Identification Test. ISSL: Stress Symptom Inventory for Adults.

**Table 2 healthcare-11-00954-t002:** Generalized linear models comparing the adjusted effect size of each cardiovascular risk factor (independent variables) on the total number of risk factors (dependent variable) after adjustment by age and sex.

Variables	ß (Raw)	ß (Stand.)	[95%CI]	*p* Value
AUDIT score	0.554	0.110	[0.060; 0.162]	<0.001 *
Smoking load	0.478	0.028	[0.007; 0.049]	0.009 *
ISSL sumscore	0.118	0.021	[0.006; 0.037]	0.008 *
Body mass index	0.050	0.016	[0.008; 0.023]	<0.001 *
Glycemia	0.078	0.002	[0.000; 0.004]	0.047 *
Mean blood pressure	0.000	0.000	[−0.003; 0.003]	0.994
AUDIT score	0.554	0.110	[0.060; 0.162]	<0.001 *
Smoking load	0.478	0.028	[0.007; 0.049]	0.009 *

AUDIT: Alcohol Use Problems Identification Test. ISSL: Stress Symptom Inventory for Adults. 95%CI: confidence interval at the 95% level. * Statistical evidence of significance at *p* < 0.05.

**Table 3 healthcare-11-00954-t003:** Generalized linear models comparing adjusted effect size of cardiovascular disease risk factors (independent variables) on musculoskeletal function (dependent variable) after adjustment by age and sex.

Variables	ß (Raw)	ß (Stand.)	[95%CI]	*p* Value
Handgrip strength, %				R^2^ = 0.132
ISSL sumscore	−0.003	−0.003	[−0.013; 0.008]	0.570
Smoking load	0.000	0.000	[−0.001; 0.001]	0.987
AUDIT score	0.000	0.000	[−0.006; 0.006]	0.981
Mean blood pressure	0.007	0.019	[0.001; 0.037]	0.048 *
Glycemia	0.000	0.000	[0.000; 0.000]	0.574
Body mass index	0.006	0.002	[−0.004; 0.008]	0.508
Maximal inspiratory pressure, %				R^2^ = 0.272
ISSL sumscore	−0.002	−0.001	[−0.012; 0.010]	0.822
Smoking load	0.003	0.001	[0.000; 0.002]	0.145
AUDIT score	0.013	0.005	[−0.001; 0.011]	0.125
Mean blood pressure	0.013	0.025	[0.006; 0.043]	0.013 *
Glycemia	0.000	0.000	[0.000; 0.000]	0.799
Body mass index	0.024	0.006	[0.000; 0.012]	0.054
Maximal expiratory pressure, %				R^2^ = 0.194
ISSL sumscore	0.002	0.002	[−0.010; 0.015]	0.743
Smoking load	0.003	0.001	[0.000; 0.002]	0.198
AUDIT score	0.020	0.009	[0.002; 0.016]	0.013 *
Mean blood pressure	0.002	0.005	[−0.016; 0.028]	0.629
Glycemia	−0.001	0.000	[0.000; 0.000]	0.458
Body mass index	0.025	0.008	[0.000; 0.015]	0.035 *

AUDIT: Alcohol Use Problems Identification Test. ISSL: Stress Symptom Inventory for Adults. 95%CI: confidence interval at the 95% level.; * Statistical evidence of significance at *p* < 0.05.

## Data Availability

The dataset and scripts for statistical analysis can be obtained upon request by the principal investigator.

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
