# Peer review of "Association of Multiple Cardiovascular Risk Factors with Musculoskeletal Function in Acute Coronary Syndrome Ward Inpatients"

_healthcare, 2023, doi:10.3390/healthcare11070954_

Round 1
Reviewer 1 Report
This study identifies a connection between musculoskeletal function and coronary artery disease. Through clinical measurements at patient intake, the authors have found that weaker musculoskeletal function, specifically hand grip strength and maximal inspiratory and expiratory pressures, correlate with coronary artery disease. The manuscript is well written, and data presented accordingly. However, some improvements are suggested. Please see below.
Major edits:
Title: The title does not make sense as written. Is the word “and” missing in-between “Factors” and “Musculoskeletal”? Did 6 authors miss this?
Lines 68-70: You aptly mention the reason for the study, but do not go into detail as to what the differences in the design and population were in comparison to previous studies. This would go a long way to set up what you propose to do and set as a standard of practice going forward.
Lines 256-259: This sentence is confusing. The findings of this study do not support what?
Lines 245-246 and lines 300-301: How do you conflate these two sentences? Does this mean that skeletal muscle strength does not associate with MBP, but it does associate with hypertension, obesity and drinking behaviors and therefore should be considered as a sign of CAD? If so, this needs to be more clearly stated.
Figure 1: Figure panels need more information on the Y axis. Describe what those numbers mean (percentage, packs of cigarettes a day, etc.). For example, Mean blood pressure (mmHg).
Figure 1: The figure legend needs more than a title. Describe your data! What did you find? Describe any statistics you preformed. Note what the X axis means (what does 1-5 mean). All this data belongs in the figure legend and the authors should not rely on the reader to find this information in the body of the text.
Lines 279-280: I am glad that you listed a major limitation of the study. While reading this submitted article, I thought about the cause-effect of skeletal muscle weakness. Could inspiratory and expiratory pressures correlate with smoking and not necessarily muscle strength and thus more to do with lung function that CAD? If so, this should be addressed.
Minor edits:
· BMI is not defined in the text.
· MBP is not defined in the text.
Author Response
March 10, 2023
Response to Reviewer #1
- This study identifies a connection between musculoskeletal function and coronary artery disease. Through clinical measurements at patient intake, the authors have found that weaker musculoskeletal function, specifically hand grip strength and maximal inspiratory and expiratory pressures, correlate with coronary artery disease. The manuscript is well written, and data presented accordingly. However, some improvements are suggested. Please see below.
Thank you for your positive assessment of our manuscript. We strongly believe your comments improved the overall quality of the manuscript. Hope it is a great contribution to the management of patients with coronary artery disease. All changes made in the manuscript (insertions or corrected information) were highlighted in red color to show Editors and Reviewers where changes have been made.
Major edits:
- Title: The title does not make sense as written. Is the word “and” missing in-between “Factors” and “Musculoskeletal”? Did 6 authors miss this?
Thank you for noticing this missing word. Edited as suggested.
- Lines 68-70: You aptly mention the reason for the study, but do not go into detail as to what the differences in the design and population were in comparison to previous studies. This would go a long way to set up what you propose to do and set as a standard of practice going forward.
Thank you for this suggestion. The differences in study design and populations were only briefly mentioned in the Introduction section because we detailed them in the Discussion section. We agree though that more information may contribute to understanding our choice of study design and population. The text in Introduction was revised to:
Introduction
In adults with ACS, reports on the association between musculoskeletal function and CVD risk are also controversial and are difficult to compare due to major differences in study design (cross-sectional [16–19]; longitudinal [20,21]; Mendelian randomization [22,23]) and populations (men and women separately [16]; men only [17]; women only [18]; general population in a surgical intensive care unit [20]; cases of coronary artery disease/myocardial infarction and controls [21–23]).
Discussion
Cross-sectional studies conducted in the general population reported an inverse association between HGS/BMI and systolic blood pressure among United States adults (men and women separately [16]), in Taiwan adults (men only [17]), and Japanese adults (women only [18]). Another cross-sectional study in Chinese elderlies found a low discriminative power of HGS/BMI (or HGS/weight) on several risk factors for CVD—including hypertension and diabetes mellitus—as well as to presenting ³1 CVD risk factor [19]. A longitudinal analysis of patients admitted to a general population surgical intensive care unit reported Medical Research Council scores but not HGS predicted length of stay, but only 1 (0.9%) patient had ACS as an admission diagnosis [20]. Another longitudinal study showed a reduced mortality risk with higher HGS univariately and after adjustment for age, gender, and other cardiovascular risk factors incusing BMI, type-2 diabetes mellitus, hypertension. history of smoking [21]. A Mendelian randomization study of cases with coronary artery disease/myocardial infaction and controls showed that each 1-kg increase in HGS decreased CAD risk by 6%; the study also reported no significant association was found for type 2 diabetes, BMI and fasting glucose [22]. Another Mendelian randomization study showed that both observational and genetically predicted low handgrip strength was associated with high all-cause and particularly cardiovascular mortality after adjustment for age, sex, and phenotypes for diabetes, body mass index among others [23].
- Lines 256-259: This sentence is confusing. The findings of this study do not support what?
Thank you for pointing this out. The sentence was revised to:
Discrepancies among these findings might also be explained by different methods being used, e.g., ‘absolute’ or ‘relative’ handgrip strength. However, the findings of this study using prediction equations for HGS do not support such discrepancies.
- Lines 245-246 and lines 300-301: How do you conflate these two sentences? Does this mean that skeletal muscle strength does not associate with MBP, but it does associate with hypertension, obesity and drinking behaviors and therefore should be considered as a sign of CAD? If so, this needs to be more clearly stated.
Thank you for your comment. We apologize for the imprecision of our original report. From Table 2, we found HGS was not associated with the total number of CVD risk factors (ß = 0.000 [-0.003; 0.003], P = 0.994). From Table 3, we found that HGS was associated with MBP in a model with other surrogate measures of CVD risk after adjustment by age and sex (ß = 0.019 [0.001; 0.037], P = 0.048). Also from Table 3, we found that MIP% was also associated with MBP (ß = 0.025 [0.006; 0.043], P = 0.013) and MEP% was associated with both AUDIT score (ß = 0.009, P = 0.013) and BMI (ß = 0.008, P = 0.035). We then appreciate the opportunity to revise our writing on these findings and further discuss them as follows:
Discussion
The major findings suggest that peripheral muscle strength is directly associated with mean blood pressure, whereas respiratory muscle strength is directly associated with mean blood pressure, alcohol drinking, and body mass index in adults hospitalized for ACS.
Peripheral muscle strength showed evidence of association with all surrogate measures of CVD risk factors (ISSL sumscore, smoking load, AUDIT score, glycemia, BMI) but MBP.
Respiratory muscle strength was found positively associated with some surrogate measures of CVD risk factors (MBP, AUDIT score, and BMI).
Our findings corroborate this interactive nature as all (ISSL sumscore, smoking load, AUDIT score, glycemia, BMI) but one (MBP) surrogate measure of cardiovascular risk factors was positively associated with the total number of CVD risk factors (Table 2). Such lack of association of MPB may be explained by the majority of patients (n = 60, 87%) reporting hypertension, which is a leading cause of ACS.
Conclusions
Peripheral muscle strength is directly associated with mean blood pressure, whereas respiratory muscle strength is directly associated with mean blood pressure, alcohol drinking, and body mass index in adults hospitalized for ACS. These findings reinforce the association of skeletal muscle functional status with comorbidities (e.g. hypertension, obesity) and health behaviors (drinking habits). Skeletal muscle strength measured at cardiology ward admission should be interpreted in the context of its association with CVD risk factors in this population.
- Figure 1: Figure panels need more information on the Y axis. Describe what those numbers mean (percentage, packs of cigarettes a day, etc.). For example, Mean blood pressure (mmHg).
Thank you for your suggestion. Measurement units were added to the Y-axis of each plot.
- Figure 1: The figure legend needs more than a title. Describe your data! What did you find? Describe any statistics you preformed. Note what the X axis means (what does 1-5 mean). All this data belongs in the figure legend and the authors should not rely on the reader to find this information in the body of the text.
Thank you for your suggestion. The legend for Figure 1 was updated as follows:
Figure 1. Boxplots of independent surrogate measures of cardiovascular disease risk factors (ISSL sumscore, smoking load, AUDIT score, mean blood pressure, glycemia, body mass index) across the combination of risk factors (stress, smoking habits, alcohol drinking, hypertension, diabetes mellitus, obesity). The horizontal axis represents the total number of cardiovascular risk factors (1-5) identified per participant based on the clinical and/or laboratory assessment. Notice no patient presented with all 6 cardiovascular risk factors simultaneously.
- Lines 279-280: I am glad that you listed a major limitation of the study. While reading this submitted article, I thought about the cause-effect of skeletal muscle weakness. Could inspiratory and expiratory pressures correlate with smoking and not necessarily muscle strength and thus more to do with lung function that CAD? If so, this should be addressed.
Thank you for your insight regarding this point. We revised the Discussion section to introduce this issue:
For instance, to what extent do inspiratory and expiratory pressures correlate with smoking and not necessarily muscle strength – and thus more to do with pulmonary diseases than to CAD itself – should be further investigated.
Minor edits:
- BMI is not defined in the text.
Body mass index was defined the first time it appeared in the text.
- MBP is not defined in the text.
Thanks for this input. Mean blood pressure (MBP) is now defined.

Reviewer 2 Report
dear authors,
it is an interesting topic. the paper is well described although I have some suggestions: the tables need to have more descriptive legends, with the siglas; te discussion needs to be more details with more references; the conclusion can have the impact of this study in clinical performance.
best regards
Author Response
March 10, 2023
Response to Reviewer #2
- It is an interesting topic. the paper is well described although I have some suggestions:
Thank you for your positive assessment of our manuscript. We strongly believe your comments improved the overall quality of the manuscript. Hope it is a great contribution to the management of patients with coronary artery disease. All changes made in the manuscript (insertions or corrected information) were highlighted in red color to show Editors and Reviewers where changes have been made.
- The tables need to have more descriptive legends, with the siglas;
Thank you for your suggestion. Tables and figures legends were more updated as follows:
Table 1. Descriptive analysis of the sample of patients hospitalized for acute coronary syndrome in the cardiology ward.
Variable |
Description |
Values |
Sample size, n (%) |
|
69 |
|
Female |
23 (33%) |
|
Male |
46 (67%) |
Age, years |
|
55 ± 6 |
GRACE risk score, n (%) |
|
107 ± 23 |
|
Low risk |
32 (46%) |
|
Intermediate risk |
24 (35%) |
|
High risk |
13 (19%) |
Clinical/laboratory exams |
|
|
|
Heart rate, b/min |
74 ± 12 |
|
Systolic blood pressure, mmHg |
122 ± 16 |
|
Diastolic blood pressure, mmHg |
76 ± 10 |
|
Pulse pressure, mmHg |
46 ± 15 |
|
Mean pressure, mmHg |
92 ± 10 |
|
Blood saturation, % |
97 ± 1 |
|
Glycemia, mg/dL |
135 ± 54 |
Anthropometry |
|
|
|
Body height, m |
1.62 ± 0.09 |
|
Body mass, kg |
72.0 ± 12.8 |
|
Body mass index, kg/m2 |
27.4 ± 3.9 |
|
Abdominal circumference, cm |
97.0 ± 12.2 |
Nutritional status, n (%) |
|
|
|
Thin |
1 (1%) |
|
Eutrophic |
20 (29%) |
|
Overweight |
33 (48%) |
|
Obese I |
13 (19%) |
|
Obese II |
2 (3%) |
Body composition |
|
|
|
Body fat, % |
21 ± 7 |
|
Fat mass, kg |
24 ± 9 |
|
Thin mass, kg |
48 ± 10 |
Health behaviors |
|
|
Smoking load, pack-years |
|
16.1 ± 21.0 |
AUDIT score, n (%) |
|
|
|
Probable dependency |
3 (4%) |
|
High risk |
4 (6%) |
|
Medium risk |
13 (19%) |
|
Low risk |
49 (71%) |
ISSL, Phase I, n (%) |
|
|
|
Alert |
13 (19%) |
|
No alert |
56 (81%) |
ISSL, Phase II, n (%) |
|
|
|
Resistant |
42 (61%) |
|
No resistant |
27 (39%) |
ISSL, Phase III, n (%) |
|
|
|
Exhaustion |
13 (19%) |
|
No exhaustion |
56 (81%) |
Risk factors for cardiovascular disease, n (%) |
|
|
|
1 |
11 (16%) |
|
2 |
16 (23%) |
|
3 |
17 (25%) |
|
4 |
19 (28%) |
|
5 |
6 (9%) |
Risk factors, n (%) |
|
|
|
Hypertension |
60 (87%) |
|
Stress |
43 (62%) |
|
Obesity |
31 (45%) |
|
Smoking |
30 (43%) |
|
Drinking |
21 (30%) |
|
Diabetes mellitus |
15 (22%) |
Length of stay, days |
|
40 ± 26 |
Musculoskeletal function |
|
|
Handgrip strength |
|
|
|
Dominant hand, kg |
29 ± 10 |
|
Dominant hand, predict % |
74 ± 21 |
Respiratory muscle strength |
|
|
|
Maximal inspiratory pressure, cmH2O |
-67 ± 31 |
|
Maximal expiratory pressure, cmH2O |
61 ± 28 |
|
Maximal inspiratory pressure, predict % |
65 ± 27 |
|
Maximal expiratory pressure, predict % |
57 ± 22 |
GRACE: Global Registry of Acute Coronary Events. AUDIT: Alcohol Use Problems Identification Test. ISSL: Stress Symptom Inventory for Adults.
Table 2. Generalized linear models comparing the adjusted effect size of each cardiovascular risk factor (independent variables) on the total number of risk factors (dependent variable) after adjustment by age and sex.
Variables |
ß (Raw) |
ß (Stand.) |
[95%CI] |
P value |
AUDIT score |
0.554 |
0.110 |
[0.060; 0.162] |
<0.001* |
Smoking load |
0.478 |
0.028 |
[0.007; 0.049] |
0.009* |
ISSL sumscore |
0.118 |
0.021 |
[0.006; 0.037] |
0.008* |
Body mass index |
0.050 |
0.016 |
[0.008; 0.023] |
<0.001* |
Glycemia |
0.078 |
0.002 |
[0.000; 0.004] |
0.047* |
Mean blood pressure |
0.000 |
0.000 |
[-0.003; 0.003] |
0.994 |
AUDIT score |
0.554 |
0.110 |
[0.060; 0.162] |
<0.001* |
Smoking load |
0.478 |
0.028 |
[0.007; 0.049] |
0.009* |
AUDIT: Alcohol Use Problems Identification Test. ISSL: Stress Symptom Inventory for Adults. 95%CI: confidence interval at the 95% level.
* Statistical evidence of significance at P < 0.05
Table 3. Generalized linear models comparing adjusted effect size of cardiovascular disease risk factors (independent variables) on musculoskeletal function (dependent variable) after adjustment by age and sex.
Variables |
ß (Raw) |
ß (Stand.) |
[95%CI] |
P value |
Handgrip strength, % |
|
|
|
R2 = 0.132 |
ISSL sumscore |
-0.003 |
-0.003 |
[-0.013; 0.008] |
0.570 |
Smoking load |
0.000 |
0.000 |
[-0.001; 0.001] |
0.987 |
AUDIT score |
0.000 |
0.000 |
[-0.006; 0.006] |
0.981 |
Mean blood pressure |
0.007 |
0.019 |
[0.001; 0.037] |
0.048* |
Glycemia |
0.000 |
0.000 |
[0.000; 0.000] |
0.574 |
Body mass index |
0.006 |
0.002 |
[-0.004; 0.008] |
0.508 |
Maximal inspiratory pressure, % |
|
|
|
R2 = 0.272 |
ISSL sumscore |
-0.002 |
-0.001 |
[-0.012; 0.010] |
0.822 |
Smoking load |
0.003 |
0.001 |
[0.000; 0.002] |
0.145 |
AUDIT score |
0.013 |
0.005 |
[-0.001; 0.011] |
0.125 |
Mean blood pressure |
0.013 |
0.025 |
[0.006; 0.043] |
0.013* |
Glycemia |
0.000 |
0.000 |
[0.000; 0.000] |
0.799 |
Body mass index |
0.024 |
0.006 |
[0.000; 0.012] |
0.054 |
Maximal expiratory pressure, % |
|
|
|
R2 = 0.194 |
ISSL sumscore |
0.002 |
0.002 |
[-0.010; 0.015] |
0.743 |
Smoking load |
0.003 |
0.001 |
[0.000; 0.002] |
0.198 |
AUDIT score |
0.020 |
0.009 |
[0.002; 0.016] |
0.013* |
Mean blood pressure |
0.002 |
0.005 |
[-0.016; 0.028] |
0.629 |
Glycemia |
-0.001 |
0.000 |
[0.000; 0.000] |
0.458 |
Body mass index |
0.025 |
0.008 |
[0.000; 0.015] |
0.035* |
AUDIT: Alcohol Use Problems Identification Test. ISSL: Stress Symptom Inventory for Adults. 95%CI: confidence interval at the 95% level.
* Statistical evidence of significance at P < 0.05
- The discussion needs to be more details with more references;
Thank you for your comment. The Discussion was revised according to your and other reviewers’ suggestions. Particularly, we added the following updated references in this section:
Another longitudinal study showed a reduced mortality risk with higher HGS univariately and after adjustment for age, gender, and other cardiovascular risk factors incusing BMI, type-2 diabetes mellitus, hypertension. history of smoking [21]. A Mendelian randomization study of cases with coronary artery disease/myocardial infaction and controls showed that each 1-kg increase in HGS decreased CAD risk by 6%; the study also reported no significant association was found for type 2 diabetes, BMI and fasting glucose [22]. Another Mendelian randomization study showed that both observational and genetically pre- dicted low handgrip strength was associated with high all-cause and particularly cardiovascular mortality after adjustment for age, sex, and phenotypes for diabetes, body mass index among others [23].
[21] Larcher B, Zanolin-Purin D, Vonbank A, Heinzle CF, Mader A, Sternbauer S, et al. Usefulness of Handgrip Strength to Predict Mortality in Patients With Coronary Artery Disease. Am J Cardiol 2020;129:5–9. https://doi.org/10.1016/j.amjcard.2020.05.006.
[22] Xu L, Hao YT. Effect of handgrip on coronary artery disease and myocardial infarction: a Mendelian randomization study. Sci Rep 2017;7:954. https://doi.org/10.1038/s41598-017-01073-z.
[23] Park S, Lee S, Kim Y, Lee Y, Kang MW, Kim K, et al. Relation of Poor Handgrip Strength or Slow Walking Pace to Risk of Myocardial Infarction and Fatality. Am J Cardiol 2022;162:58–65. https://doi.org/10.1016/j.amjcard.2021.08.061.
- The conclusion can have the impact of this study in clinical performance.
Thank you for your suggestion. We opt to keep the clinical impact of our findings in the Discussion section for a more in-depth discussion at this section and to keep Conclusions concise. We thought updated the text in Discussion and Conclusions using the new references as follows:
Discussion
The clinical implications of the current study findings comprise that knowing the CVD risk profile might allow planning interventions for adults with ACS while staying in a cardiology ward. With new evidence suggesting a causal relationship between HGS and all-cause and cardiovascular mortality [22,23], it is expected that these findings might also contribute to primary-to-secondary prevention for further guidance on changing health behaviors and controlling for comorbidities that can lead to an increase in both the expectation and quality of life in this population as well as to avoid hospital readmissions [2].
Conclusions
Peripheral muscle strength is directly associated with mean blood pressure, whereas respiratory muscle strength is directly associated with mean blood pressure, alcohol drinking, and body mass index in adults hospitalized for ACS. These findings reinforce the association of skeletal muscle functional status with comorbidities (e.g. hypertension, obesity) and health behaviors (drinking habits). Skeletal muscle strength measured at cardiology ward admission should be interpreted in the context of its association with CVD risk factors in this population.

Reviewer 3 Report
Dear Authors,
Thank you for the opportunity to review your paper.
I think that it is an interesting topic and you've reported interesting data.
I have just some suggestions in order to improve it.
Abstract
Are you sure that the background is correct? It looks more like an aim fo the study.
Introduction
Pag. 2 line 61-62: please clarify this sentence.
Sincerely, reading the background about muscle strength is not clear if you would like to measure its correlation with other risk factors or itself as a risk factor for CVD. Please, is it possible to answer to this question?
Material and Methods
Pag. 2 line 89: "all procedures were performed", procedures or assessments? Please clarify.
Discussion and Conclusions
I think that in the discussion and moreover in the conclusions, you should answer (or clarify) to my question based on the Introduction section. At end of the paper it should be clear the aim of the study, if the study provide new insights on this topic and the suggested research development.
Author Response
March 10, 2023
Response to Reviewer #3
- Dear Authors, Thank you for the opportunity to review your paper. I think that it is an interesting topic and you've reported interesting data. I have just some suggestions in order to improve it.
Thank you for your positive assessment of our manuscript. We strongly believe your comments improved the overall quality of the manuscript. Hope it is a great contribution to the management of patients with coronary artery disease. All changes made in the manuscript (insertions or corrected information) were highlighted in red color to show Editors and Reviewers where changes have been made.
Abstract
- Are you sure that the background is correct? It looks more like an aim fo the study.
Thank you for your comment. Headings were removed as per the journal’s Instructions for Authors.
Introduction
- 2 line 61-62: please clarify this sentence.
Thank you for your suggestion. The text was revised to:
Muscle weakness is also a known complication acquired by critically ill patients that have associated poor outcomes such as longer length of stay and more days of mechanical ventilation [9].
- Sincerely, reading the background about muscle strength is not clear if you would like to measure its correlation with other risk factors or itself as a risk factor for CVD. Please, is it possible to answer to this question?
Thank you for your comment. We rephrased our study question to:
Particularly, to what extent musculoskeletal function (i.e., peripheral or respiratory muscle strength) is associated with comorbidities and lifestyle – either as independent CVD risk factors or a clustered group of comorbidities – in patients admitted in the cardiology ward for ACS remains unknown.
Material and Methods
- 2 line 89: "all procedures were performed", procedures or assessments? Please clarify.
Thank you for your comment. We rephrased this sentence to:
All assessments were performed by the same examiner.
Discussion and Conclusions
- I think that in the discussion and moreover in the conclusions, you should answer (or clarify) to my question based on the Introduction section. At end of the paper it should be clear the aim of the study, if the study provide new insights on this topic and the suggested research development.
We thank the reviewer for these suggestions. We have revised several parts of the Discussion and Conclusions sections considering your comments above as well as those from other reviewers. Please find below major changes in these sections:
- Discussion
This study explored the association of multiple CVD risk factors with musculoskeletal function in adults hospitalized for ACS. The major findings suggest that peripheral muscle strength is directly associated with mean blood pressure, whereas respiratory muscle strength is directly associated with mean blood pressure, alcohol drinking, and body mass index in adults hospitalized for ACS. Major strengths comprise using valid, reliable instruments to evaluate the CVD risk factors and respiratory and peripheral muscle strength. Also, both demographics and risk factors profile of this cohort are similar to other studies in adults hospitalized for ACS [38,39], highlighting the external validity of our findings.
Peripheral muscle strength showed evidence of association with all surrogate measures of CVD risk factors (ISSL sumscore, smoking load, AUDIT score, glycemia, BMI) but MBP. Reports on the association between HGS and CVD risk are also controversial and are difficult to compare herein due to major differences in designs and populations. Cross-sectional studies conducted in the general population reported an inverse association between HGS/BMI and systolic blood pressure among United States adults (men and women separately [16]), in Taiwan adults (men only [17]), and Japanese adults (women only [18]). Another cross-sectional study in Chinese elderlies found a low discriminative power of HGS/BMI (or HGS/weight) on several risk factors for CVD—including hypertension and diabetes mellitus—as well as to presenting ³1 CVD risk factor [19]. A longitudinal analysis of patients admitted to a general population surgical intensive care unit reported Medical Research Council scores but not HGS predicted length of stay, but only 1 (0.9%) patient had ACS as an admission diagnosis [20]. Another longitudinal study showed a reduced mortality risk with higher HGS univariately and after adjustment for age, gender, and other cardiovascular risk factors incusing BMI, type-2 diabetes mellitus, hypertension. history of smoking [21]. A Mendelian randomization study of cases with coronary artery disease/myocardial infaction and controls showed that each 1-kg increase in HGS decreased CAD risk by 6%; the study also reported no significant association was found for type 2 diabetes, BMI and fasting glucose [22]. Another Mendelian randomization study showed that both observational and genetically pre- dicted low handgrip strength was associated with high all-cause and particularly cardiovascular mortality after adjustment for age, sex, and phenotypes for diabetes, body mass index among others [23]. Discrepancies among these findings might also be explained by different methods being used, e.g., ‘absolute’ or ‘relative’ handgrip strength. However, the findings of this study using prediction equations for HGS do not support such discrepancies.
Respiratory muscle strength was found positively associated with some surrogate measures of CVD risk factors (MBP, AUDIT score, and BMI). Such a relationship appears paradoxical as smoking [10], hypertension [11], and diabetes mellitus [12] have been inversely related to respiratory muscle function. The relationship with obesity seems still debatable; MIP and MEP were reported as not different across classifications of nutritional status by BMI [13] and as positively correlated with BMI [14,15]. The relationship of respiratory muscle strength with stress or alcohol drinking remains not investigated, although evidence points to a direct association between either risk factors and mortality [40,41]. It is worth noting that such relationships were reported independently for each CVD risk factor in different populations; using self-reported, non-standard, or different instruments for assessing CVD risk factors; and analyzing raw and/or predict functional outcomes with or without adjustment for possible covariates. The reasoning underlying the above-mentioned direct association remains uncertain. Given that CVD are multifactorial, the interactive, complex nature of their risk factors has been investigated in adults with ACS [42,43]. Our findings corroborate this interactive nature as all (ISSL sumscore, smoking load, AUDIT score, glycemia, BMI) but one (MBP) surrogate measure of cardiovascular risk factors was positively associated with the total number of CVD risk factors (Table 2). Such lack of association of MPB may be explained by the majority of patients (n = 60, 87%) reporting hypertension, which is a leading cause of ACS. Accordingly, the majority of participants (28%) showed four risk factors, similar to other cohorts reported in Brazil [38] and other countries [39]. Future studies might investigate whether other factors might explain this direct association, including moderation and/or mediation effects between CVD risk factors and respiratory muscle strength.
The major limitation of this study is the cross-sectional design that precludes inference about the cause-effect associations for the observed models. Another major limitation is that CVD risk factors were operationalized with a single measure, e.g., a point measurement of blood pressure and fasting glycemia, that might express the current condition rather than the trait for a risk profile. Also, data on additional CVD risk factors (e.g.,physical activity) and functional outcomes (e.g., pulmonary function, functional exercise capacity) were not collected and might provide additional insights into the relationships investigated herein. For instance, to what extent do inspiratory and expiratory pressures correlate with smoking and not necessarily muscle strength – and thus more to do with pulmonary diseases than to CAD itself – should be further investigated. Moreover, the possible confounding effect of drug treatment other than antihypertensive or hypoglycemic agents was not considered. Finally, even though national prediction equations were used, nationwide regional differences¾mainly on anthropometry and nutritional status¾might help explain the low predicted values for the musculoskeletal functions observed herein.
The clinical implications of the current study findings comprise that knowing the CVD risk profile might allow planning interventions for adults with ACS while staying in a cardiology ward. With new evidence suggesting a causal relationship between HGS and all-cause and cardiovascular mortality [22,23], it is expected that these findings might also contribute to primary-to-secondary prevention for further guidance on changing health behaviors and controlling for comorbidities that can lead to an increase in both the expectation and quality of life in this population as well as to avoid hospital readmissions [2].
- Conclusions
Peripheral muscle strength is directly associated with mean blood pressure, whereas respiratory muscle strength is directly associated with mean blood pressure, alcohol drinking, and body mass index in adults hospitalized for ACS. These findings reinforce the association of skeletal muscle functional status with comorbidities (e.g. hypertension, obesity) and health behaviors (drinking habits). Skeletal muscle strength measured at cardiology ward admission should be interpreted in the context of its association with CVD risk factors in this population.
